# Continuous Measurement in Neurocritical Care of Cerebral Blood Flow (CBF) Calculated from ICP and Central Venous Pressure

**DOI:** 10.3390/neurolint17040049

**Published:** 2025-03-25

**Authors:** Erik Ryding

**Affiliations:** Department of Clinical Neurophysiology, University Hospital of Lund, 222 41 Lund, Sweden; erk@ryding.net

**Keywords:** CBF, ICP, central venous pressure, venous resistance, *rICP*

## Abstract

**Background/Objectives:** In neurocritical care, usually, the only continuous measurement of brain pathophysiology is intracranial pressure (ICP). The objective of this study was to find the relationship between cerebral blood flow (CBF) and parameters usually measured in neurocritical care, mainly central venous pressure and ICP. **Methods:** If the venous outflow of the CBF is considered, the CBF is controlled only by two parameters, the *rICP* (the ICP minus the venous blood pressure in the venous sinus at its outflow) and the *Rv* (the flow resistance of the soft-walled veins). For the *rICP*, the sinus blood pressure can be calculated from the central venous pressure (measured at the same horizontal level as the ICP) and the cervical venous flow resistance. For the *Rv*, the systolic ICP increase indicates the systolic arterial inflow volume, which then flows out before the diastole. The mean ICP increase divided by the mean outflow of the increased blood volume gives the *Rv*. This method of calculating the CBF by dividing the *rICP* by the *Rv* was named **CBF(1)**. For validation of **CBF(1)**, data from nine subjects in an open study were used. The data were ICP and MR blood flow measurements of arterial inflow and jugular vein outflow. Since the *rICP, Rv*, and CBF were unknown, an iterative method was needed to calculate these parameters. **Results:** The observed *Rv* and *rICP* values showed a close correlation, which indicated that CBF was dependant on the *rICP* only. Consequently, the comparison between the data in the study of the nine subjects, and the calculated values from **CBF(1)**, boiled down to a comparison between the supine ICP values and the calculated rICP. The comparison showed that the rICP and supine ICP had highly significant similarity, and that the **CBF(1)** method was validated. **Conclusions:** A method for CBF measurement from ICP data in neurocritical care was found.

## 1. Background

Neurocritical care is intensive care of subjects with serious brain conditions, like trauma, bleeding, infection, and oedema, sometimes combined with other serious physical damage.

The special feature of the brain that makes observation difficult is its enclosure in the scull bone and hard meninges. X-ray and MR of the scull and brain can give important information, but cannot be used continuously.

Continuous measurement of intracranial pressure is one obvious solution, but unfortunately, it has traditionally been measured against the air pressure, which makes the measured intracranial pressure a reliable physiological parameter only when the patient is in the supine position.

Cerebral blood flow brings oxygen to and removes carbon dioxide and other waste products from the intracranial space. It is a vital parameter that would be highly clinically useful if it could be measured continuously.

The aim of this article [1] is to present a method for continuous CBF measurement calculated from parameters that are routinely measured in neurocritical care, like intracranial pressure and central venous pressure.

## 2. Methods

If the venous outflow of the CBF is considered, the flow rate can be calculated as the quotient between the driving pressure and the venous flow resistance, *Rv*.

The driving pressure for the cerebral venous outflow is the intracranial pressure, ICP, subtracted from the venous pressure in the inflow into the venous sinus (which are hard-walled venous blood vessels in the hard meninges). The driving pressure for the venous outflow is denoted as referential ICP, *rICP*.

The sinus blood pressure is estimated from the central venous pressure measured at the same horizontal level as the intracranial pressure, with correction for pressure loss at the cervical venous resistance.

The flow resistance for the venous CBF outflow, *Rv*, can be found from the increase in intracranial pressure during heart systole, which is caused by arterial inflow of new blood volume intracranially. Since the intracranial volume is constant, the inflow is matched by an equal venous blood volume outflow in the distal part of the soft-walled venous bed. The compression of the distal venous bed causes an increase in *rICP* [2]. Calculating the incoming blood volume from the *rICP* increase enables estimation of the corresponding mean outflow rate during the pulse stroke. The mean *rICP* increase divided by the mean outflow rate during a pulse stroke gives the *Rv*. The mean *rICP* divided by the *Rv* then gives the CBF; this method is named **CBF(1)**.

### 2.1. Validation

For validation of the described **CBF(1)** method, measurements of ICP and the systolic inflow of blood volume from patients in neurointensive care were needed. We found these data in an open study [3] where we could freely use the data provided; we have declared the source. The study was performed on volunteers who were in improving or stable condition, and it was approved by the Ethical Committee.

The Unnerbäck study contained ten subjects (the data from one could not be used, due to a faulty ICP measurement). The diagnoses ranged from head trauma to encephalitis and brain tumour. The available parameters were ICP and arterial inflow and venous outflow velocities (measured by MR). The CBF or *rICP* were not known.

The arterial inflow was measured for all four arteries, but the venous outflow was only measured from the jugular veins and scaled to the arterial inflow during the pulse wave. Since the patients at measurement were in the supine position, the venous measurements did not contain the outflow caused by the limited intracranial space, which goes through the occipital veins.

### 2.2. Calculations

Assuming a CBF value, the venous outflow during a pulse wave is matched to the arterial inflow. The difference in arterial and venous flow velocities (initially higher arterial flow velocity) causes an increase in intracranial blood volume, which will eventually disappear at diastole. By scaling and fitting the ICP curve during the blood volume curve, we obtain the scale factor between the intracranial volume increase and ICP increase, which gives the mean *rICP* for the pulse interval.The **CBF(1)** method described above gives the mean *Rv* for the pulse interval.The quotient between the mean *rICP* and mean *Rv* gives a new CBF value which can replace the originally assumed one; this can be used in further calculations.

This iterative method was necessary, since there was mutual interdependence between the *rICP*, *Rv*, and CBF. The convergence was rapid, with the same CBF value at the beginning and at the end, after 6–8 iterations or fewer.

## 3. Results

The iterative calculations gave the *rICP*, *Rv*, and CBF for all nine subjects. It was evident that the CBF values closely equalled the value obtained from the *rICP* divided by the *Rv*. This means that the obtained CBF vales were the maximum possible values.

Another important feature was a close correlation between the logarithms Ln(*rICP*) and Ln(*Rv*), with the effect that CBF was a function of the *rICP* only. This gave the maximum possible CBF:***CBFmax*** = *rICP*^−0.80^ × 444 (mL/(100 g × min))
which is a valid method for CBF if the arteriolar sphincters are maximally dilated, and the CBF is controlled by the *Rv* only.

It also gave the relationship between the *Rv* and *rICP*:*Rv* = (*rICP*)^1.80^ × 0.0135 (mmHg/(mL/s))
which is the venous flow resistance that controls the CBF when the arteriolar sphincters are maximally dilated. Since the CBF was a function of the *rICP* only, the comparison between the data of [3] and the results of the iterative calculations using the **CBF(1)** method boiled down to a comparison between the *rICP* and supine ICP values. In the supine position, the ICP and rICP values are expected to be very similar, since the venous blood pressure difference between the brain and the heart should be close to zero. Two subjects had a much lower ICP than rICP (as expected when sitting up). The others had a highly significant correlation between the *rICP* and ICP along the identity line. The **CBF(1)** method was validated.

## 4. Implications

The *rICP* is an important new parameter (the difference between the ICP and the cerebral venous outflow pressure) which can replace ICP in CBF calculations. The *rICP* is (in contrast to the ICP) a physiological parameter, and is independent of head or body position.

Continuous CBF: The **CBF(1)** method, as described in the Methods Section, works if the scull is intact, and if the *rICP* is high enough that the systolic inflow of arterial blood gives a measurable *rICP* increase. This gives the possibility of making an algorithm that continuously calculates the CBF from the inflow of *rICP* values. The **CBF(max)** (which is a simpler method) works if the venous resistance (coupled to the *rICP*) is high enough that the arteriolar sphincters must be maximally dilated (for optimal tissue perfusion). Like **CBF(1)**, the **CBF(max)** can be calculated continuously. It was a surprising finding that for subjects in improving condition, the CBF followed the **CBF(max)** down to about 10 mmHg.

*Other parameters*: The *Rv* (the venous blood flow resistance, dependent on the *rICP*) is essential for calculation of CBF, especially at increased *rICP*. To our knowledge, the dependence of the *Rv* on the *rICP* (or ICP) has not been measured in humans before.

*General comment:* Continuous CBF constitutes another parameter besides ICP that can be used to examine the vital status of a damaged brain. Also, the new parameters besides CBF, the *Rv* and *rICP*, can be calculated continuously, and together, they contribute to a better understanding of cerebral pathophysiology. This study offers a new way to study CBF, from the venous outflow side.

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
