# Peer review of "Continuous Measurement in Neurocritical Care of Cerebral Blood Flow (CBF) Calculated from ICP and Central Venous Pressure"

_2035-8377, 2025, doi:10.3390/neurolint17040049_

Round 1

Reviewer 1 Report (Previous Reviewer 1)

Comments and Suggestions for Authors

This is interesting commentary. No further revisions required

Author Response

Comments and Suggestions for Author: This is interesting commentary. No further revisions required.

Answer: Thank you. No revisions are made.

Reviewer 2 Report (New Reviewer)

Comments and Suggestions for Authors

The presented material highlights the main results of a previously published work, https://doi.org/10.1038/s41598-024-74983-4 (where a noteworthy approach to calculating cerebral blood flow based on data measured during neurocritical care was introduced). The possibilities and prospects for using the results of that work are discussed.

The material is well structured, well designed and can be recommended for publication. The only thing that is missing, in my opinion, is the key figures from the original article (and without them it is very difficult to perceive the text). But perhaps this was done on purpose to motivate the reader to look at the figures there (and at the same time read the entire article).

Author Response

Comments and Suggestions for Author: The material is well structured, well designed and can be recommended for publication. The only thing that is missing, in my opinion, is the key figures from the original article (and without them it is very difficult to perceive the text). But perhaps this was done on purpose to motivate the reader to look at the figures there (and at the same time read the entire article).

Answer: Thank you. I quite agree that the Comment is a pointer to the original article which should be read as a whole.

Reviewer 3 Report (New Reviewer)

Comments and Suggestions for Authors

The artıcle is well documented. I think it will be useful to clinicians.

Acceptable

Author Response

Comments and Suggestions for Author: The article is well documented. I think it will be useful to clinicians. Acceptable.

Answer: Thank you. It is intended as pointer to the Original Article to make it available for clinicians.

This manuscript is a resubmission of an earlier submission. The following is a list of the peer review reports and author responses from that submission.

Round 1

Reviewer 1 Report

Comments and Suggestions for Authors

Interesting study, but first of all this is not a commentary but a short research paper. It should be extended, supplemented with figures and submitted in this category. Otherwise it is very difficult to comprehend and verify

Author Response

Question: This is not a commentary but a short research paper. It should be extended, supplemented with figures and submitted in this category. Otherwise it is very difficult to to comprehend and verify.

Answer: This is indeed (as stated) a commentary on my article in Scientific Reports 2024 ,14, 23268. The article is in a general scientific magazine, and the commentary is intended medical (neurology) readers as a pointer to its existence. Without adequate knowledge in the field, and without ability to read the article, the commentary is unavoidable very difficult to comprehend and verify.

Reviewer 2 Report

Comments and Suggestions for Authors

The commentary discusses a recently published article in "Scientific Reports" by the same author, who proposed an alternative method for calculating cerebral blood flow (CBF). The article explores the relationship between venous perfusion, referential intracranial pressure (rICP), and venous outflow resistance (Rv). Traditional techniques for measuring CBF, such as X-ray and MRI, can only be performed intermittently and do not allow continuous monitoring. This new method enables continuous measurement of CBF using parameters commonly assessed in neurocritical care, such as intracranial and central venous pressure. The commentary also validates this innovative CBF measurement approach. Additionally, the author highlights that the relationship between Rv and rICP has not previously been measured in humans. Beyond CBF, Rv, and rICP, other parameters can enhance our understanding of cerebral pathophysiology.

Author Response

Comment: This new method enables continuous measurement of CBF using parameters commonly assessed in neurocritical care, such as intracranial and central venous pressure. 

Answer: This is of cause the key point of my Commentary. I thank you for a careful and adequate review.

Reviewer 3 Report

Comments and Suggestions for Authors

Hello

Thanks for submitting the paper to this journal. Though it is an interesting attempt to calculate CBF but the article needs to be redrafted by a native English speaking author. Also the format of the draft including the abstract is not in the expected format of a research article and needs a thorough revision.

Further comments you could find here:

1. In abstract, the conclusion cannot be a new method is discovered! It needs to be framed with regards to the objective of the study.

2. The methods need to be more elaborate and scientific mentioning the classical comparator arm with the new method.

3.  The baseline characteristics of study population with etiology needs to be mentioned.

Thanks

Comments on the Quality of English Language

Hello

Thanks for submitting the paper to this journal. Though it is an interesting attempt to calculate CBF but the article needs to be redrafted by a native English speaking author. Also the format of the draft including the abstract is not in the expected format of a research article and needs a thorough revision.

Author Response

Comment 1. In abstract, the conclusion cannot be a new method is discovered! It needs to be framed with regards to the objective of the study.

Answer 1: The object of the study was to discover a new method to calculate CBF from ICP and CVP measurements.

Comment 2: The methods need to be more elaborate and scientific mentioning the classical comparator arm with the new method.

Answer 2: The article, which is commented on, is published in a magazine for general science since it was difficult to read for reviewers in the adequate medical one. I have written a comment to my research paper as a pointer for medical (neurology) readers to find. Highly qualified scientific reviewers have thoroughly examine the article's scientific value and found it adequate.

Comment 3: The baseline characteristics of study population with etiology needs to be mentioned.

Answer 3: The etiology of the study population is adequately described in the the article which is commented on. 

Round 2

Reviewer 1 Report

Comments and Suggestions for Authors

After revision I found this work appropriate for publication

Comments on the Quality of English Language

It is OK

Reviewer 3 Report

Comments and Suggestions for Authors

Hello,

thanks for revised version submission.

The commentary to an original article needs to be drafted in the format of commentary and not in the format of original article.

iThenticate report shows 13% similarity to the original article.

The commentary should not only be original but also talk about it's implications in real world practice. 

Thanks

Comments on the Quality of English Language

The draft needs a though revision both from the linguistic as well as grammatical view point. Thanks